# An Improved RAIM Availability Assessment Method Based on the Characteristic Slope

**DOI:** 10.3390/s24113283

**Published:** 2024-05-21

**Authors:** Jing Zhao, Dan Song, Jitao Wang

**Affiliations:** 1China Transport Telecommunications and Information Center, Beijing 100011, China; zhaojing0070@163.com; 2Graduate School, Beihang University, Beijing 100191, China; 3School of Electronic Information Engineering, Beihang University, Beijing 100191, China; wangjitao@buaa.edu.cn

**Keywords:** RAIM, availability, characteristic slope, protection level

## Abstract

The availability assessment is an important step for onboard application in Receiver Autonomous Integrity Monitoring (RAIM)s. It is commonly implemented using the protection level (PL)-based method. This paper analyzes the deficiencies of three kinds of PL-based methods: RAIM availability might be optimistically or conservatively assessed using the classic-PL-base method; might be conservatively assessed using the enhanced-PL-based method, and neither be optimistically nor conservatively assessed using the ideal-PL-based method with the cost of large calculation amount on-board. An improved slope-based RAIM availability assessment method is proposed, in which the characteristic slope is designed as the assessment basis, and its threshold that can exactly match the integrity risk requirement is derived. The slope-based method has the same RAIM availability assessment result as the ideal-PL-based method. Moreover, because the slope threshold can be calculated offline and searched online, the on-board calculation burden can be reduced using the slope-based method. Simulation is presented to verify the theoretical analysis of the RAIM availability assessment performances for the three PL-based and the slope-based methods.

## 1. Introduction

The integrity of the Global Navigation Satellite System (GNSS) is one of the important factors to ensure civil aviation safety. There are three categories of GNSS integrity augmentation systems: Satellite-Based Augmentation System (SBAS), Ground-Based Augmentation System (GBAS) and Aircraft-Based Augmentation System (ABAS). The first two categories are at the system level, and the latter category is at the user level [1]. ABAS can be implemented with Receiver Autonomous Integrity Monitoring (RAIM), which provides a navigation solution with guaranteed integrity by consistency checking among measurements [2].

For an onboard application, RAIM is executed in two steps: the RAIM availability assessment and the satellite fault detection [1]. The former (RAIM availability assessment) is used to assess in advance whether the navigation solution can meet the integrity risk requirement with the fault detection procedure. For decades, RAIM availability assessment has been achieved by calculating the protection level (PL), the upper bound of the position error corresponding to the integrity risk requirement [3]. The threshold of the PL is the alert limit (AL), the upper bound of the user-allowed position error. If the PL is lower than the AL, RAIM is considered available; otherwise, it is considered unavailable.

Many studies have focused on determining how to obtain a lower PL to improve the availability of RAIM. Some of these studies were devoted to developing new navigation solution calculation methods, for example, the improved Integrity-Optimized RAIM (NIORAIM) [4] and the optimal weighted average solution (OWAS) [5] methods used for the snapshot RAIM algorithm. These methods can obviously decrease the PL with a slight increase in nominal position error. In addition, some studies have committed to accurately modeling the stochastic measurement noise, such as the discrete error-distribution (NavDEN) model proposed by Rife and Pervan [6] and the distribution model considering both elevation angle and orbit type proposed by Fan [7]. These measurement noise models are all helpful for obtaining a tight PL. 

However, in the process of pursuing a lower PL, i.e., higher availability of RAIM, there is a key issue that is ignored by most researchers: whether the PL can accurately assess RAIM availability. Milner and Ochieng noted this issue [3]. They qualitatively described that the classic PL, the product of the characteristic slope and the Minimum Detection Bias (MDB) proposed by Brown and Chin [8], was too optimistic for RAIM availability assessment. The slope is a geometric feature-related parameter that qualitatively describes the relationship between positioning error and pseudo-range residual [9]. The reason is “PL < AL” might not mean that the integrity risk satisfies the requirement for the measurement bias less than MDB. Meanwhile, its enhancement, abbreviated as the enhanced PL in the following, which provides an additional term to protect against the variation in position error proposed by Angus [10], is too conservative. The reason is “PL ≥ AL” might not mean that the integrity risk exceeds the requirement. Here, measurement bias means the measurement error caused by the satellite fault, which is different from the measurement noise in the nominal mode. Furthermore, they proposed the ideal PL. It is the minimum PL value that guarantees the integrity risk, satisfying its requirement for arbitrary measurement bias. The ideal PL can prevent RAIM availability assessment from being optimistic or conservative. However, it cannot be solved analytically. A numerical search for the ideal PL begins with an improbably large value [3], which leads to a large amount of calculation, increasing the computational burden of a GNSS receiver or an onboard computer. 

In recent years, most researchers have focused on Advanced RAIM (ARAIM), in which PL is calculated after fault detection [11]. ARAIM supports multi-constellation dual-frequency GNSS integrity monitoring. Multiple hypothesis solution separation (MHSS) algorithm is used in ARAIM. How to solve the accurate PL for MHSS is a research hotspot, including the PL calculation method for each fault mode and the optimization strategy for the ultimate PL [12,13,14,15,16]. Jiang and Wang adopted the ideal PL [3] in ARAIM and verified it was more accurate than other PLs for the ARAIM availability assessment [17,18]. ARAIM is still in the theoretical research stage and is not currently being applied in engineering practice. 

Compared with ARAIM, RAIM has two deficiencies. The first is that RAIM is designed for a single constellation, monitoring only a single satellite fault [2]. ARAIM is designed for double constellations, monitoring not only the single satellite fault but also the multiple satellite faults and the constellation fault [12]. The second is classic, and the enhanced PLs are not rigorous enough for RAIM availability assessment, while the PL of ARAIM is much more rigorous. However, the on-board calculation of RAIM using the classic or the enhanced PLs is much less than that of ARAIM. For a single constellation, RAIM can still be used, but there is a problem needs to be considered, founding a RAIM availability assessment method both satisfying the rigor and maintaining the low on-board computational burden.

In this paper, a slope-based RAIM availability assessment method is proposed to solve the above problem. The characteristic slope is taken as the assessment basis. Using the ideal slope threshold, this method can achieve a consistent RAIM availability assessment with the ideal-PL-based method. The ideal slope threshold can be calculated offline and searched online because it is only related to one geometric parameter. 

The remainder of this paper is organized as follows: Section 2 states some technical backgrounds. Section 3 reviews PL-based RAIM availability assessment methods, including the classic, enhanced and ideal PLs. The deficiencies of the classic and enhanced-PL-based methods can be analyzed quantitatively using the rates of optimistic or conservative assessment. Section 4 proposes the slope-based RAIM availability assessment method after the derivation of the ideal slope threshold. Section 5 gives an overview of the simulation results of the classic-PL-based, enhanced-PL-based, ideal-PL-based and slope-based methods. Section 6 concludes this work with a brief summary. The discussion of this paper takes vertical integrity as an example, using one kind of classic RAIM snapshot algorithm, i.e., the least squares residuals (LSR) algorithm. In this paper, the measurement noise is assumed to be independent white Gaussian noise (WGN).

## 2. Technical Background

Before discussing the RAIM availability assessment methods, some technical backgrounds need to be stated, including the derivation of the integrity risk requirement for the single-satellite fault mode, the definition of the vertical characteristic slope and the specific meaning of RAIM being available.

### 2.1. Integrity Risk Requirement for the Single-Satellite Fault Mode

The integrity risk P(HMI) [19] is the probability of undetected faults causing unacceptably large errors in the estimated position [20]. HMI is short for hazardous misleading information (HMI). P(HMI) can be divided into three fault modes, the nominal mode, the single-satellite fault mode and the multiple-satellite fault mode, expressed as: (1)$$P(HMI)=\sum_{i=0}^{2}P(HMI,iF)=\sum_{i=0}^{2}P(HMI\left|iF\right.)P(iF)$$

In Equation (1), 0F, 1F and 2F respectively represent the nominal, the single-satellite fault and the multiple-satellite fault modes, where P(iF) is the prior probability of the fault mode iF and $P(HMI\left|iF\right.)$ is the probability of HMI under the iF fault mode. $P(0F)=(1-P_{sat}{)}^{K}$ and $P(1F)=C_{K}^{1}P_{sat}(1-P_{sat}{)}^{K}$, where $C_{K}^{1}$ means the number of combinations for choosing one element from K elements; $P_{sat}$ is the prior fault probability of a satellite, and *K* is the total visible satellite number. Taking vertical plane for example, $P(HMI\left|iF\right.)$ is calculated with the following equation [21]:(2)$$P(HMI\left|iF\right.)=P(\left|VPE\right|\geq VAL\left|iF\right.)P(Ts<Td\left|iF\right.)$$

In Equation (2), VPE is the vertical position error; VAL is the vertical AL; Ts and Td are the fault detection test statistic and threshold, respectively. 

To ensure the integrity of the navigation system, P(HMI) should be less than its requirement, denoted as $P_{r}(HMI)$.
(3)$$P(HMI,0F)+P(HMI,1F)+P(HMI,2F)<P_{r}(HMI)$$

Given a geometry between the user and all-in-view satellites, VAL, α and $P_{sat}$, P(HMI,0F) can be calculated following: (4)$$P(HMI,0F)=(1-\alpha )[1-\int_{-VAL}^{VAL}g(x;0,a_{v}^{2})dx](1-P_{sat}{)}^{K}$$

In Equation (4), α is the allowable false alarm probability under the nominal fault mode, satisfying $P(Ts<Td\left|0F\right.)=1-\alpha$; $g(x;0,a_{v}^{2})$ is the probability density function (PDF) of the normal-distributed VPE under the nominal fault mode with mean value 0 and standard deviation value $a_{v}$, where $a_{v}$ is explained in Appendix A.

According to GNSS Evolutionary Architecture Study (GEAS) report [22], the integrity risk requirement allocated on the multiple satellites fault, denoted as $P_{r}(HMI,2F)$, can be set $1.3\times 10^{-8}$ [22].

To ensure Equation (3) to be true, the integrity risk for single fault mode should satisfy:(5)$$P(HMI,1F)<P_{r}(HMI\left|1F\right.)$$
where $P_{r}(HMI\left|1F\right.)=[P_{r}(HMI)-P_{r}(HMI,2F)-P(HMI,0F)]/P(1F)$. Thus, the problem of $P(HMI)<P_{r}(HMI)$ has evolved into the problem of $P(HMI\left|1F\right.)<P_{r}(HMI\left|1F\right.)$.

Moreover, to ensure that $P_{r}(HMI,1F)$ is nonnegative, the threshold for $a_{v}$, denoted as $Ta_{v}$, can be derived from the inequality $P_{r}(HMI)-P_{r}(HMI,2F)-P(HMI,0F)\geq 0$.
(6)$$Ta_{\nu }=-VAL/\Phi^{-1}\left(\frac{P_{r}(HMI)-P_{r}(HMI,2F)}{2(1-\alpha )(1-P_{sat}{)}^{K}}\right)$$
where $\Phi^{-1}\left(\cdot \right)$ represents the inverse function of the cumulative distribution function (CDF) for the standard normal distribution. $a_{v}\geq Ta_{v}$ indicates that only the integrity risk of the nominal and multiple-satellite fault modes have exceeded the total requirement, i.e., $R_{0}+P_{r}(HMI,2)\geq P_{r}(HMI)$. 

### 2.2. Vertical Characteristic Slope

The vertical characteristic slope is defined according to these two parameters [2],
(7)$$Slope_{m}=\sqrt{\frac{a_{3m}^{2}}{s_{mm}}}$$
where $a_{3m}$ and $s_{mm}$ respectively characterizes VPE and Ts change caused by the measurement bias of the *m*-th visible satellite, signed as $VS_{m}$, is the faulty satellite. The details of $a_{3m}$ and $s_{mm}$ can be seen in Appendix A and Appendix B, respectively. For a specific bias value, a faulty satellite with a large slope value will present a high $P(HMI\left|1F\right.)$, and a faulty satellite with a small slope value will present a low $P(HMI\left|1F\right.)$. The “characteristic slope” will be abbreviated as “slope” hereafter.

### 2.3. Specific Meaning of RAIM Being Available

RAIM being available refers to $P(HMI)<P_{r}(HMI)$, i.e., $P(HMI\left|1F\right.)<P_{r}(HMI\left|1F\right.)$ for the arbitrary measurement bias value while RAIM being unavailable refers to $P(HMI\left|1F\right.)\geq P_{r}(HMI\left|1F\right.)$ for at least one measurement bias.

A specific example is used to intuitively explain the meaning of RAIM being available. The 32-satellite GPS constellation is used in this example. The pseudorange measurement is assumed to be the dual-frequency ionosphere-free combination of L1 and L5. The standard deviation of the measurement noise for $VS_{i}$, signed as $\sigma_{i}$, is set according to the ARAIM interim report [11]. For the location of 37° N latitude, 117° E longitude and height 0 m and the epoch of UTC 14 March 2019 17:15:00, there are 9 visible satellites with a masking angle of 10°. Their vertical slope values are recorded in Table 1.

Figure 1 presents the base-10 logarithm of $P(HMI\left|1F\right.)$ for PRN5, PRN6 and PRN13 with the measurement bias $\xi_{b}$ in the interval of (0. 30 m). These three satellites have the top three vertical slope values, as shown in Table 1. The $\xi_{b}-\mathrm{lg}P(HMI\left|1F\right.)$ curves for these three satellites follow the same order as the slope values, which illustrates that a faulty satellite with a large slope will present a high $P(HMI\left|1F\right.)$. The $\xi_{b}-\mathrm{lg}P(HMI\left|1F\right.)$ curves for the other six visible satellites must be lower than that of PRN 13 because their vertical slopes are smaller. In this example, RAIM would be unavailable if PRN5 was the faulty satellite because $P(HMI\left|1F\right.)$ is larger than $P_{r}(HMI\left|1F\right.)$ for $\xi_{b}$ values in the range from 11 m to 15 m. Therefore, the intersection between the $\xi_{b}-\mathrm{lg}P(HMI\left|1F\right.)$ curve of the faulty satellite and the $P_{r}(HMI\left|1F\right.)$ line means that RAIM is unavailable. RAIM would be available if the faulty satellite was one of the other visible satellites except for PRN5, because its $P(HMI\left|1F\right.)$ smaller than $P_{r}(HMI\left|1F\right.)$ at an arbitrary measurement bias. Therefore, the separation between the $\xi_{b}-\mathrm{lg}P(HMI\left|1F\right.)$ curve of the faulty satellite and the $P_{r}(HMI\left|1F\right.)$ line means that RAIM is available.

Because the faulty satellite is unknown in actual situations, RAIM is considered available only if $P(HMI\left|1F\right.)<P_{r}(HMI\left|1F\right.)$ is true for the arbitrary measurement bias in the worst case, i.e., the satellite with the maximum slope being faulty. 

## 3. PL-Based RAIM Availability Assessment

The classic PL, enhanced PL, and ideal PL are reviewed in this section.

### 3.1. Classic PL and Its Enhancement

The classic PL, denoted as $PL_{c}$, is defined as the product of the maximum slope $Slope_{max}$ and the minimum detectable bias $\sqrt{\lambda_{a}}$ as follows: (8)$$PL_{c}={Slope}_{max}\sqrt{\lambda_{a}}$$
where $\lambda_{a}$ is the noncentral parameter of the fault detection test statistic Ts under single-satellite fault mode [23].

The enhanced PL, denoted as $PL_{e}$, has an additional term that protects against variation in the random error of the position solution on the basis of $PL_{c}$ as follows [10]:(9)$$PL_{e}={Slope}_{max}\sqrt{\lambda_{a}}+\alpha (P_{MD})\sigma$$
where $\alpha (P_{MD})=\Phi^{-1}(1-P_{MD})$ and σ is the standard deviation of the position error distribution.

Slope reflects the relationship between the position error and the pseudorange residual. PL is the projection of the pseudorange residual on the position error as shown in Equations (8) and (9). According to $\lambda ={\left(\frac{\xi_{b}}{\sigma_{m}^{2}}\right)}^{2}s_{mm}$ derived in Appendix B, the measurement bias $\xi_{b}$ for a faulty satellite $VS_{m}$, which causes Ts to obey $\chi^{2}(K-4,\lambda_{a})$, is $\xi_{b}=\sigma_{m}^{2}\sqrt{\frac{\lambda_{a}}{s_{mm}}}$. Based on the position error derivation in Appendix A, this $\xi_{b}$ makes the VPE under single-satellite fault mode obey:(10)$$VPE∼N(Slope_{m}\sqrt{\lambda_{a}},a_{v}^{2})$$

Contrasting Equation (8) with Equation (10), the classic vertical PL, denoted as $VPL_{c}$, is the expectation of VPE at $\xi_{b}=\sigma_{m}^{2}\sqrt{\frac{\lambda_{a}}{s_{mm}}}$ with the maximum slope. Therefore, $P(\left|VPE\right|\geq VPL_{c}\left|1F\right.)=0.5$ at $\xi_{b}=\sigma_{m}^{2}\sqrt{\frac{\lambda_{a}}{s_{mm}}}$ in the worst case. Similarly, the enhanced vertical PL, denoted as $VPL_{e}$, is the upper quantile of the $P_{MD}$ of VPE distribution at $\xi_{b}=\sigma_{m}^{2}\sqrt{\frac{\lambda_{a}}{s_{mm}}}$ with the maximum slope. Therefore, $P(\left|VPE\right|\geq VPL_{e}\left|1F\right.)=P_{MD}$ at $\xi_{b}=\sigma_{m}^{2}\sqrt{\frac{\lambda_{a}}{s_{mm}}}$ in the worst case.

### 3.2. Deficiency of Classic- and Enhanced-PL-Based RAIM Availability Assessment

Here defines H_0_ as “$\forall \xi_{b}$,$P(HMI\left|1F\right.)<P_{r}(HMI\left|1F\right.)$” and H_1_ as “$\exists \xi_{b}$,$P(HMI\left|1F\right.)\geq P_{r}(HMI\left|1F\right.)$” in the worst case. Therefore “$VPL<VAL\left|H_{1}\right.$” or “$VPL\geq VAL\left|H_{0}\right.$” respectively mean the RAIM availability assessment is optimistic or conservative.

For $\forall \xi_{b}$
(11)$$\begin{array}{c} VPL<VAL\Leftrightarrow P\left(\left|VPE\right|\geq VPL\left|1F\right.\right)P\left(Ts<Td\left|1F\right.\right)>\\ P(\left|VPE\right|\geq VAL\left|1F\right.)P(Ts<Td\left|1F\right.)\\ \Leftrightarrow P_{a}(HMI\left|1F\right.)>P(HMI\left|1F\right.) \end{array}$$
and
(12)$$VPL\geq VAL\Leftrightarrow P_{a}(HMI\left|1F\right.)\leq P(HMI\left|1F\right.)$$
where $P(\left|VPE\right|\geq VPL\left|1F\right.)P(Ts<Td\left|1F\right.)=P_{a}(HMI\left|1F\right.)$, 

Similar to $P(HMI\left|1F\right.)$, $P_{a}(HMI\left|1F\right.)$ changes with the $\xi_{b}$ value. Moreover, the value of $P_{a}(HMI\left|1F\right.)$ is known at $\xi_{b}=\sigma_{m}^{2}\sqrt{\frac{\lambda_{a}}{s_{mm}}}$, $P_{a}(HMI\left|1F\right.)=0.5P_{MD}$ for $VPL_{c}$ and $P_{a}(HMI\left|1F\right.)=P_{MD}^{2}$ for $VPL_{e}$. A $\xi_{b}-\mathrm{lg}P_{a}(HMI\left|1F\right.)$ curve can be generated, which is higher than the $\xi_{b}-\mathrm{lg}P(HMI\left|1F\right.)$ curve with VPL<VAL and lower than the $\xi_{b}-\mathrm{lg}P(HMI\left|1F\right.)$ curve with VPL≥VAL. The positional relationship among the $\xi_{b}-\mathrm{lg}P(HMI\left|1F\right.)$ curve, the $\xi_{b}-\mathrm{lg}P_{a}(HMI\left|1F\right.)$ curve, and the $\mathrm{lg}P_{r}(HMI\left|1F\right.)$ line jointly determine whether the PL-based RAIM availability assessment is conservative or optimistic. The following is a RAIM availability assessment discussion according to the positional relationship between the $\xi_{b}-\mathrm{lg}P_{a}(HMI\left|1F\right.)$ curve and the $\mathrm{lg}P_{r}(HMI\left|1F\right.)$ line.

#### 3.2.1. Separation

In this situation, there are three kinds of positional relationships among the $\xi_{b}-\mathrm{lg}P(HMI\left|1F\right.)$ curve, the $\xi_{b}-\mathrm{lg}P_{a}(HMI\left|1F\right.)$ curve and the $\mathrm{lg}P_{r}(HMI\left|1F\right.)$ line, as shown in Figure 2.

The $\xi_{b}-\mathrm{lg}P(HMI\left|1F\right.)$ curve intersects with the $\mathrm{lg}P_{r}(HMI\left|1F\right.)$ line, and is higher than the blue dashed $\xi_{b}-\mathrm{lg}P_{a}(HMI\left|1F\right.)$ curve, shown as the brown $P_{1}(HMI\left|1F\right.)$ curve in Figure 2. In this situation, RAIM is unavailable and the VPL is larger than the VAL, i.e., $VPL\geq VAL\left|H_{1}\right.$, meaning a successful detection of “RAIM being unavailable”.

The $\xi_{b}-\mathrm{lg}P(HMI\left|1F\right.)$ curve is separated from the $\mathrm{lg}P_{r}(HMI\left|1F\right.)$ line, and is higher than the blue dashed curve $\xi_{b}-\mathrm{lg}P_{a}(HMI\left|1F\right.)$ curve, shown as the black curve $P_{2}(HMI\left|1F\right.)$ in Figure 2. In this situation, RAIM is available but the VPL is larger than the VAL, i.e., $VPL\geq VAL\left|H_{0}\right.$, meaning a conservative assessment.

The $\xi_{b}-\mathrm{lg}P(HMI\left|1F\right.)$ curve is separated from the $\mathrm{lg}P_{r}(HMI\left|1F\right.)$ line, and is lower than the blue dashed $\xi_{b}-\mathrm{lg}P_{a}(HMI\left|1F\right.)$ curve, shown as the blue curve in Figure 2. In this situation, RAIM is available, and the VPL is smaller than the VAL, i.e., $VPL<VAL\left|H_{0}\right.$, meaning a successful detection of “RAIM being available”.

Therefore, for the condition that the $\xi_{b}-\mathrm{lg}P_{a}(HMI\left|1F\right.)$ curve is separated from the $\mathrm{lg}P_{r}(HMI\left|1F\right.)$ line, the RAIM availability might be conservatively assessed.

#### 3.2.2. Intersection

In this situation, there are also three kinds of positional relationships among the $\xi_{b}-\mathrm{lg}P(HMI\left|1F\right.)$ curve, the $\xi_{b}-\mathrm{lg}P_{a}(HMI\left|1F\right.)$ curve and the $\mathrm{lg}P_{r}(HMI\left|1F\right.)$ line, as shown in Figure 3.

The $\xi_{b}-\mathrm{lg}P(HMI\left|1F\right.)$ curve intersects with the $\mathrm{lg}P_{r}(HMI\left|1F\right.)$ line, and is higher than the blue dashed $\xi_{b}-\mathrm{lg}P_{a}(HMI\left|1F\right.)$ curve, shown as the brown $P_{1}(HMI\left|1F\right.)$ curve in Figure 3 In this situation, RAIM is unavailable and the VPL is larger than the VAL, i.e., $VPL\geq VAL\left|H_{1}\right.$, meaning a successful detection of “RAIM being unavailable”.

The $\xi_{b}-\mathrm{lg}P(HMI\left|1F\right.)$ curve intersects with the $\mathrm{lg}P_{r}(HMI\left|1F\right.)$ line, and is lower than the blue dashed $\xi_{b}-\mathrm{lg}P_{a}(HMI\left|1F\right.)$ curve, shown as the black $P_{2}(HMI\left|1F\right.)$ curve in Figure 3. In this situation, RAIM is unavailable, but the VPL is smaller than the VAL, i.e., $VPL<VAL\left|H_{1}\right.$, meaning an optimistic assessment.

The $\xi_{b}-\mathrm{lg}P(HMI\left|1F\right.)$ curve is separated from the $\mathrm{lg}P_{r}(HMI\left|1F\right.)$ line, and is lower than the blue dashed $\xi_{b}-\mathrm{lg}P_{a}(HMI\left|1F\right.)$ curve, shown as the blue curve $P_{3}(HMI\left|1F\right.)$ in Figure 3. In this situation, RAIM is available, and the VPL is smaller than the VAL, i.e., $VPL<VAL\left|H_{0}\right.$, meaning the successful detection of “RAIM being available”.

Therefore, for the condition that the $\xi_{b}-\mathrm{lg}P_{a}(HMI\left|1F\right.)$ curve intersects with the $\mathrm{lg}P_{r}(HMI\left|1F\right.)$ line, the RAIM availability might be optimistically assessed.

#### 3.2.3. Tangency

In this situation, there are two kinds of positional relationships among the $\xi_{b}-\mathrm{lg}P(HMI\left|1F\right.)$ curve, the $\xi_{b}-\mathrm{lg}P_{a}(HMI\left|1F\right.)$ curve and the $\mathrm{lg}P_{r}(HMI\left|1F\right.)$ line, as shown in Figure 4.

The $\xi_{b}-\mathrm{lg}P(HMI\left|1F\right.)$ curve intersects with the $\mathrm{lg}P_{r}(HMI\left|1F\right.)$ line, and is higher than the blue dashed $\xi_{b}-\mathrm{lg}P_{a}(HMI\left|1F\right.)$ curve, shown as the brown $P_{1}(HMI\left|1F\right.)$ curve in Figure 4. In this situation, RAIM is unavailable, and the VPL is larger than the VAL, i.e., $VPL\geq VAL\left|H_{1}\right.$, meaning a successful detection of “RAIM being unavailable”.

The $\xi_{b}-\mathrm{lg}P(HMI\left|1F\right.)$ curve is separated from the $\mathrm{lg}P_{r}(HMI\left|1F\right.)$ line, and is lower than the blue dashed $\xi_{b}-\mathrm{lg}P_{a}(HMI\left|1F\right.)$ curve, shown as the black $P_{2}(HMI\left|1F\right.)$ curve in Figure 4. In this situation, RAIM is available and the VPL is smaller than the VAL, i.e., $VPL<VAL\left|H_{0}\right.$, meaning a successful detection of “RAIM being available”.

Therefore, for the condition that the $\xi_{b}-\mathrm{lg}P_{a}(HMI\left|1F\right.)$ curve is tangent to the $\mathrm{lg}P_{r}(HMI\left|1F\right.)$ line, both optimistic and conservative assessments can be prevented.

Based on the above analysis, the accuracy of PL-based RAIM availability assessment depends on the positional relationship between the $\xi_{b}-\mathrm{lg}P_{a}(HMI\left|1F\right.)$ curve and the $\mathrm{lg}P_{r}(HMI\left|1F\right.)$ line. There is a risk of conservative assessment when the $\xi_{b}-\mathrm{lg}P_{a}(HMI\left|1F\right.)$ curve is separated from the $\mathrm{lg}P_{r}(HMI\left|1F\right.)$ line and risk of optimistic assessment when the $\xi_{b}-\mathrm{lg}P_{a}(HMI\left|1F\right.)$ curve intersects with the $\mathrm{lg}P_{r}(HMI\left|1F\right.)$ line. PL-based RAIM availability assessment is accurate only if the $\xi_{b}-\mathrm{lg}P_{a}(HMI\left|1F\right.)$ curve is tangent to the $\mathrm{lg}P_{r}(HMI\left|1F\right.)$ line. Because only the $P_{a}(HMI\left|1F\right.)$ value at a specific $\xi_{b}$, i.e., $\xi_{b}=\sigma_{m}^{2}\sqrt{\frac{\lambda_{a}}{s_{mm}}}$, is determined for $VPL_{c}$ or $VPL_{e}$, the position relationship between the entire $\xi_{b}-\mathrm{lg}P_{a}(HMI\left|1F\right.)$ curve and the $\mathrm{lg}P_{r}(HMI\left|1F\right.)$ line is uncertain. Consequently, both optimistic and conservative assessments might happen when using $VPL_{c}$ or $VPL_{e}$ to assess whether vertical RAIM is available. 

Because $VPL_{c}<VPL_{e}$, the $\xi_{b}-\mathrm{lg}P_{a}(HMI\left|1F\right.)$ curve for $VPL_{c}$ is much higher than that for $VPL_{e}$. Thus the possibility of intersection between the $\xi_{b}-\mathrm{lg}P_{a}(HMI\left|1F\right.)$ curve and the $\mathrm{lg}P_{r}(HMI\left|1F\right.)$ line for $VPL_{c}$ is much higher than that for $VPL_{e}$, which may lead to an optimistic assessment, while the possibility of separation between the $\xi_{b}-\mathrm{lg}P_{a}(HMI\left|1F\right.)$ curve and the $\mathrm{lg}P_{r}(HMI\left|1F\right.)$ line for $VPL_{e}$ is much higher than that for $PL_{c}$, which may lead to conservative assessment. Consequently, the optimistic assessment risk of using $VPL_{c}$ is higher than that of using $VPL_{e}$ for vertical RAIM availability; in contrast, the conservative assessment risk of using $VPL_{e}$ is higher than that of using $VPL_{c}$. 

### 3.3. Ideal Protection Level 

According to the above analysis, the ideal positional relationship between the $\xi_{b}-\mathrm{lg}P_{a}(HMI\left|1F\right.)$ curve and the $\mathrm{lg}P_{r}(HMI\left|1F\right.)$ line is tangency, which can prevent both optimistic and conservative RAIM availability assessments. The ideal VPL, denoted as $VPL_{d}$, proposed by Milner and Ochieng, satisfies this condition. It matches the exact required integrity risk for the worst-case bias (WCB), the measurement bias presenting the highest integrity risk. Thus $VPL_{d}$ forms a $\xi_{b}-\mathrm{lg}P_{a}(HMI\left|1F\right.)$ curve tangent to the $\mathrm{lg}P_{r}(HMI\left|1F\right.)$ line. If $VPL_{d}<VAL$, the $\xi_{b}-\mathrm{lg}P(HMI\left|1F\right.)$ curve must be separated from the $\mathrm{lg}P_{r}(HMI\left|1F\right.)$ line, which means that vertical RAIM is available; otherwise the $\xi_{b}-\mathrm{lg}P(HMI\left|1F\right.)$ curve must be tangent to or intersect with the $\mathrm{lg}P_{r}(HMI\left|1F\right.)$ line, which means that vertical RAIM is unavailable.

$VPL_{d}$ is the solution of:(13)$$\max[P_{a}(HMI\left|1F\right.)]=P_{r}(HMI\left|1F\right.)$$

## 4. Slope-Based RAIM Availability Assessment

In addition to the ideal PL, there is another ideal test statistic for RAIM availability assessment: the slope. Both optimistic and conservative RAIM availability assessments can be prevented using the slope once an ideal threshold is found. The following is a deviation of this ideal threshold.

### 4.1. Derivation of the Ideal Threshold for the Slope

The ideal slope threshold derivation begins with searching for a condition satisfying $P(HMI\left|1F\right.)<P_{r}(HMI\left|1F\right.)$ for all possible measurement bias values of an arbitrary faulty satellite. To ensure $P(HMI\left|1F\right.)<P_{r}(HMI\left|1F\right.)$ constantly true, the maximum value of $P(HMI\left|1F\right.)$ should be less than $P_{r}(HMI\left|1F\right.)$,
(14)$$\max[P(HMI\left|1F\right.)]<P_{r}(HMI\left|1F\right.)$$Where:(15)$$\begin{array}{ll} P(HMI\left|1F\right| & =P(\left|VPE\right|\geq VAL\left|1F\right)P(Ts<Td\left|1F\right)\\ & =[1-\int_{-VAL}^{VAL}g(x;\frac{a_{3m}\xi_{b}}{\sigma_{m}^{2}},a_{v}^{2})dx]\int_{0}^{Td}f(y;{\left(\frac{\xi_{b}}{\sigma_{m}^{2}}\right)}^{2}s_{mm},K-4)dy \end{array}$$

Substituting Equation (7) into Equation (15),
(16)$$P(HMI\left|1F\right.)=[1-\int_{-VAL}^{VAL}g(x;\frac{a_{3m}\xi_{b}}{\sigma_{m}^{2}},a_{v}^{2})dx]\int_{0}^{Td}f(y;{\left(\frac{\xi_{b}}{\sigma_{m}^{2}}\right)}^{2}{\left(\frac{a_{3m}}{Slope_{m}}\right)}^{2},K-4)dy)$$

Setting $t=\frac{x-a_{3m}\xi_{b}/\sigma_{m}^{2}}{a_{v}}$, Equation (16) can be transformed into:(17)$$P(HMI\left|1F\right.)=[1-\int_{\frac{-VAL-\frac{a_{3m}\xi_{b}}{\sigma_{m}^{2}}}{a_{v}}}^{\frac{VAL-\frac{a_{3m}\xi_{b}}{\sigma_{m}^{2}}}{a_{v}}}g(t;0,1)dt]\int_{0}^{Td}f(y;{\left(\frac{a_{3m}\xi_{b}}{\sigma_{m}^{2}}\right)}^{2}{\left(\frac{1}{Slope_{m}}\right)}^{2},K-4)dy)$$

Setting $\mu_{s}=\frac{a_{3m}\xi_{b}}{\sigma_{m}^{2}}$, $P(HMI\left|1F\right.)$ can be taken as a function of $\mu_{s}$ as follows:(18)$$\Phi (\mu_{s})=P(HMI\left|1F\right.)=[1-\int_{\frac{-VAL-\mu_{s}}{a_{v}}}^{\frac{VAL-\mu_{s}}{a_{v}}}g(t;0,1)dt]\int_{0}^{Td}f(y;{\left(\frac{\mu_{s}}{Slope_{m}}\right)}^{2},K-4)dy)$$

When substituting Equation (18) into Equation (14), an ideal threshold must exist for $Slope_{m}$ with the given $P_{r}(HMI\left|1F\right.)$, VAL, $a_{v}$, Td and K, denoted as $T_{Slope}$, which satisfies the limit situation
(19)$$\max\left\{\Phi (\mu_{s})\left|Slope_{m}=T_{Slope}\right.\right\}=P_{r}(HMI\left|1F\right.)$$

According to Equations (18) and (19), once $P_{r}(HMI\left|1F\right.)$, VAL, Td and K are given, $T_{Slope}$ is determined by only one parameter $a_{v}$.

Figure 5 presents the numerically solved $T_{Slope}$ for $a_{v}$ values with a step of 0.001 at $P_{r}(HMI)=1\times 10^{-7}$, $P_{r}(HMI,2)=1.3\times 10^{-8}$, VAL=50 m, $P_{sat}=1\times 10^{-5}$, $\alpha =1\times 10^{-6}$ and K=7,⋯,11. As shown in Figure 5, $T_{Slope}$ decreases with the increase of $a_{v}$. Each $a_{v}-T_{Slope}$ curve exhibits a nearly constant segment at the beginning and a sharply decreasing segment at the end. The $a_{v}-T_{Slope}$ curve ends when $a_{v}$ reaches $Ta_{v}$, and the spacing between two adjacent curves obviously decreases as K increases.

### 4.2. Practical Meaning of the Ideal Slope Threshold

Here, the specific example in Section 2.3 is used to intuitively explain the practical meaning of $T_{Slope}$. The $a_{v}$ value for this example is 4.944. Given $P_{r}(HMI)=1\times 10^{-7}$, $P_{r}(HMI,2F)=1.3\times 10^{-8}$, VAL=50 m, $P_{sat}=1\times 10^{-5}$ and $\alpha =1\times 10^{-6}$, the ideal slope threshold can be numerically solved according to Equation (19), and $T_{Slope}=5.390$. Figure 6 presents the $\mu_{s}-\mathrm{lg}P(HMI|1F)$ curves for Slope=6.060 (PRN5), Slope=3.010 (PRN6) and $T_{Slope}=5.390$. P(HMI|1F) for $T_{Slope}$ is calculated according to Equation (19). The $\mu_{s}-\mathrm{lg}P(HMI|1F)$ curve for $T_{Slope}$ is tangent to the $\mathrm{lg}P_{r}(HMI\left|1F\right.)$ line. The $\mu_{s}-\mathrm{lg}P(HMI|1F)$ curve for Slope=6.060, which is larger than $T_{Slope}$, is intersected with the $\mathrm{lg}P_{r}(HMI\left|1F\right.)$ line. In contrast, the $\mu_{s}-\mathrm{lg}P(HMI|1F)$ curve for Slope=3.010, which is smaller than $T_{Slope}$, is separated from the $\mathrm{lg}P_{r}(HMI\left|1F\right.)$ line. 

It can be deduced that the slope value determines the positional relationship between the $\mu_{s}-\mathrm{lg}P(HMI|1F)$ curve and the $\mathrm{lg}P_{r}(HMI\left|1F\right.)$ line. $T_{Slope}$ forms a tangent $\mu_{s}-\mathrm{lg}P(HMI|1F)$ curve to the $\mathrm{lg}P_{r}(HMI\left|1F\right.)$ line. If the slope value of a faulty satellite is larger than $T_{Slope}$, its $\mu_{s}-\mathrm{lg}P(HMI|1F)$ curve would be intersected with the $\mathrm{lg}P_{r}(HMI\left|1F\right.)$ line, meaning that $P(HMI\left|1F\right.)\geq P_{r}(HMI\left|1F\right.)$ can be satisfied for at least one possible measurement bias value. If the slope value of a faulty satellite was smaller than $T_{Slope}$, its $\mu_{s}-\mathrm{lg}P(HMI|1F)$ curve would be separated from the $\mathrm{lg}P_{r}(HMI\left|1F\right.)$ line, meaning that $P(HMI\left|1F\right.)<P_{r}(HMI\left|1F\right.)$ can be ensured at an arbitrary measurement bias value. From the analysis of this specific example, the slope is an ideal test statistic for RAIM availability assessment, with the ideal threshold $T_{slope}$ calculated according to Equation (19). Therefore, for each observation epoch, the slope of a visible satellite can be considered as “large slope” if it was larger than $T_{slope}$ and considered as “small slope” if it was smaller than $T_{slope}$.

### 4.3. Comparison of the Ideal Slope Threshold and the Ideal Protection Level 

The functions of $T_{slope}$ and $VPL_{d}$ are identical, forming a tangent $P(HMI\left|1F\right.)$ curve to the $\mathrm{lg}P_{r}(HMI\left|1F\right.)$ line. Referring to the equation for $T_{slope}$, i.e., Equation (18), the equation for $VPL_{d}$, can be formulated as follows:(20)$$\begin{array}{cc} P_{a}(HMI\left|1F\right) & =P(\left|VPE\right|\geq VPL_{d}\left|1F\right)P(Ts<Td\left|1F\right)\\ & =[1-\int_{\frac{-VPL_{d}-\mu_{s}}{a_{\upsilon }}}^{\frac{VPL_{d}-\mu_{s}}{a_{\upsilon }}}g(t;0,1)dt]\int_{0}^{Td}f(y;{\left(\frac{\mu_{s}}{Slope_{max}}\right)}^{2},K-4)dy) \end{array}$$

Analyzing this equation, $VPL_{d}$ is determined by $a_{v}$ and $Slope_{max}$ with the given $P_{r}(HMI\left|1F\right.)$, Td and K. Therefore, $T_{slope}$ is only related to $a_{v}$ while $VPL_{d}$ is related to both $a_{v}$ and $Slope_{max}$.

Considering that the PDF of the noncentral $\chi^{2}$ distribution is too complicated, both $T_{Slope}$ and $VPL_{d}$ should be solved numerically, which will sharply increase the computation burden of a GNSS receiver or an onboard computer. Thanks to the one-to-one correspondence between $a_{v}$ and $T_{Slope}$, $T_{Slope}$ can be calculated offline for discrete $a_{v}$ values in the range from 1 to $Ta_{v}$ and saved in a receiver. It can be on-board searched from the presaved data according to the specific $a_{v}$ value. However, $VPL_{d}$ needs to be calculated online after both $a_{v}$ and $Slope_{max}$ obtained.

### 4.4. Slope-Based RAIM Availability Assessment Method

Slope-based RAIM availability assessment should be implemented in the worst case to fully prevent the integrity risk. Using slope-based method, RAIM is considered available if the maximum slope is less than the ideal threshold $T_{Slope}$; otherwise, it is considered unavailable. The specific execution process for slope-based RAIM availability assessment is presented in Figure 7.

As shown in Figure 7, the inputs are the observation matrix ***H*** and the weighted matrix ***P***. The first step is an assessment based on the total number of visible satellites *K*. RAIM is considered unavailable if K≤4 because fault detection cannot be executed with less than 5 visible satellites. For K>4, the second step is an assessment based on the $a_{v}$ value. RAIM is considered to be unavailable if $a_{v}\geq Ta_{v}$ because only the sum of P(HMI,0F) and P(HMI,2F) has exceed $P_{r}(HMI)$ when $a_{v}$ exceeds $Ta_{v}$. For $a_{v}<Ta_{v}$, the last step is an assessment based on the maximum slope $Slope_{max}$. RAIM is finally considered available if $Slop{eSlope}_{max}$.

$Ta_{v}$ can be calculated online according to Equation (6) with given VAL, $P_{sat}$,α, $P_{r}(HMI)$ and $P_{r}(HMI\left|2F\right.)$. $T_{Slope}$ should be calculated offline with discrete $a_{v}$ with a small step size for different numbers of visible satellites and saved in the GNSS receiver or the onboard computer. Assuming that the total number of visible satellites is *K* and $a_{v}$ is in the interval of $\left(a_{v1},a_{v2}\right]$, where $a_{v1}$ and $a_{v2}$ are the indexes of two adjacent discrete points $\left[a_{v1},{T_{Slope}}_{1}\right]$ and $\left[a_{v2},{T_{Slope}}_{2}\right]$ presaved for *K* visible satellites, $T_{Slope}$ corresponding to $a_{v}$ should be assigned as: (21)$$T_{Slope}=\frac{T_{Slope1}-T_{Slope2}}{a_{v1}-a_{v2}}(a_{v}-a_{v1})+T_{Slope1}$$

This value is calculated according to a linear fit for $T_{Slope}$ in the interval of $\left(a_{v1},a_{v2}\right]$. Because the $a_{v}-T_{Slope}$ curve is convex as shown in Figure 5, the assigned $T_{Slope}$ value is smaller than the real $T_{Slope}$ value, which may cause a small conservative assessment risk. However, if the $a_{v}$ step size is small enough, the assigned $T_{Slope}$ value would be nearly equal to the true value, thereby preventing the small conservative assessment risk. 

A simulation is designed to find the desirable range $a_{v}$ step. $1.2\times 10^{6}$ times of $a_{v}$ and $S\mathrm{lop}e_{max}$ for different location and epochs are collected. As shown in Table 2, the times of conservative assessment increases with step size widen. Taking into account both the amount of calculation and conservative assessment rate, 0.01~0.02 is the desirable range $a_{v}$ step because it is the maximum step size with 0 time of conservative assessment. 

It should be mentioned that the calculation amount of the ideal-PL and $T_{Slope}$ is exactly the same. The ideal PL procedure begins with an improbably large VPL of 2000 m and halves the search step by checking if the corresponding integrity risk exceeds the requirement [3]. For each step of the iteration, the integrity risk needs to be calculated for different bias values with a fixed step. This process is computationally intensive. Similarly, the $T_{Slope}$ procedure begins with an improbably large slope value of 15 and halves the search step. Compared with the ideal-PL-based method, the slope-based method separates the process of numerical iteration from on-board RAIM availability assessment, reducing the burden of on-board computing.

## 5. Simulation

To compare the performance of the PL-based and slope-based methods, the vertical RAIM availability assessment for a 32-satellite GPS constellation is simulated in worldwide (latitude 60° S~60° N and longitude 180° W~180° E) for a whole day (13 March 2019 0:00:00~24:00:00). The simulation area is meshed as the grid of 1°×1° and the simulation time step is 300 s. The masking angle is set to 10°. The dual-frequency ionosphere-free combination of L1 and L5 is assumed to be the pseudorange measurement. The standard derivation of the measurement noise is set according to the ARAIM interim report [11].

The vertical RAIM availability assessment is executed for all grid points, i.e., 121×361=43,681 grid points, using the classic-PL-based ($VPL_{c}$), the enhanced-PL-based ($VPL_{e}$), the ideal-PL-based ($VPL_{d}$) and the slope-based methods respectively at each simulation epoch. In the simulation, the parameters related to vertical RAIM availability are $P_{r}(HMI)=1\times 10^{-7}$, VAL=50m [21], $P_{r}(HMI,2)=1.3\times 10^{-8}$ [22], $P_{sat}=1\times 10^{-5}$ [11], $\alpha =1\times 10^{-6}$, and $P_{MD}=0.1\%$. The $T_{Slope}$ values are pre-calculated before the simulation with an $a_{v}$ step size of 0.01.

### 5.1. Specific Example Analysis

Three specific examples for a single grid point and a simulation epoch are chosen to show the RAIM availability assessment using the four methods in detail. Table 3 records the VPLs, $Slope_{max}$ and $T_{Slope}$ values for these specific examples. Figure 8, Figure 9, Figure 10 and Figure 11 present the real $\mu_{s}-\mathrm{lg}P(HMI\left|1F\right.)$ curve for $Slope_{max}$, the $\mu_{s}-\mathrm{lg}P(HMI\left|1F\right.)$ curve for $T_{Slope}$, and the $\mu_{s}-\mathrm{lg}P_{a}(HMI\left|1F\right.)$ curves for $VPL_{c}$, $VPL_{e}$ and $VPL_{d}$ in these examples, respectively. In Figure 8, Figure 9, Figure 10 and Figure 11, the $\mu_{s}-\mathrm{lg}P(HMI\left|1F\right.)$ curve for $T_{Slope}$ and the $\mu_{s}-\mathrm{lg}P_{a}(HMI\left|1F\right.)$ curve for $VPL_{d}$ are tangent to the $\mathrm{lg}P_{r}(HMI\left|1F\right.)$ line, while the $\xi_{b}-{\mathrm{lg}}_{a}P(HMI\left|1F\right.)$ curve for $VPL_{e}$ is separated from the $\mathrm{lg}P_{r}(HMI\left|1F\right.)$ line. The positional relationship between the $\mu_{s}-{\mathrm{lg}}_{a}P(HMI\left|1F\right.)$ curve for $VPL_{c}$ and the $\mathrm{lg}P_{r}(HMI\left|1F\right.)$ line could be either intersection or separation.

As shown in Figure 8, the real $\mu_{s}-\mathrm{lg}P(HMI\left|1F\right.)$ curve for $Slope_{max}$ intersects with the $\mathrm{lg}P_{r}(HMI\left|1F\right.)$ line, which means that RAIM is actually unavailable in the first example. The $\mu_{s}-\mathrm{lg}P_{a}(HMI\left|1F\right.)$ curve for $VPL_{c}$ is higher than the real $\mu_{s}-\mathrm{lg}P(HMI\left|1F\right.)$ curve, i.e., $VPL_{c}<VAL$, which means that RAIM is assessed to be available when using $VPL_{c}$. All of the $\mu_{s}-\mathrm{lg}P(HMI\left|1F\right.)$ curve for $T_{Slope}$, the $\mu_{s}-\mathrm{lg}P_{a}(HMI\left|1F\right.)$ curves for $VPL_{e}$ and $VPL_{d}$ are lower than the real $\mu_{s}-\mathrm{lg}P(HMI\left|1F\right.)$ curve, i.e., $Slop{e\geq Slope}_{max}$, $VPL_{e}\geq VAL$ and $VPL_{d}\geq VAL$, which means that RAIM is assessed to be unavailable when using $Slope_{max}$, $VPL_{e}$ or $VPL_{d}$. Therefore, the RAIM availability is optimistically assessed when using $VPL_{c}$ and successfully assessed using when using $Slope_{max}$, $VPL_{e}$ or $VPL_{d}$.

As shown in Figure 9, the real $\mu_{s}-\mathrm{lg}P(HMI\left|1F\right.)$ curve for $Slope_{max}$ is separated from the $\mathrm{lg}P_{r}(HMI\left|1F\right.)$ line, which means that RAIM is actually available in the second example. The $\mu_{s}-\mathrm{lg}P(HMI\left|1F\right.)$ curve for $T_{Slope}$, and the $\mu_{s}-\mathrm{lg}P_{a}(HMI\left|1F\right.)$ curves for $VPL_{c}$ and $VPL_{d}$ are higher than the real $\mu_{s}-\mathrm{lg}P(HMI\left|1F\right.)$ curve, i.e., $Slop{e<Slope}_{max}$, $VPL_{c}<VAL$ and $VPL_{d}<VAL$, which means that RAIM is assessed to be available when using $Slope_{max}$, $VPL_{c}$ or $VPL_{d}$. The $\mu_{s}-{\mathrm{lg}}_{a}P(HMI\left|1F\right.)$ curve for $VPL_{e}$ is lower than the real $\mu_{s}-\mathrm{lg}P(HMI\left|1F\right.)$ curve, i.e., $VPL_{e}\geq VAL$, which means that RAIM is assessed to be unavailable when using $VPL_{e}$. Therefore, RAIM availability is conservatively assessed when using $VPL_{e}$ and successfully assessed when using $Slope_{max}$, $VPL_{c}$ or $VPL_{d}$.

As shown in Figure 10, the real $\mu_{s}-\mathrm{lg}P(HMI\left|1F\right.)$ curve for $Slope_{max}$ is separated from the $\mathrm{lg}P_{r}(HMI\left|1F\right.)$ line, which means that RAIM is actually available in the third example. Both the $\mu_{s}-\mathrm{lg}P(HMI\left|1F\right.)$ curve for $T_{Slope}$ and the $\mu_{s}-{\mathrm{lg}}_{a}P(HMI\left|1F\right.)$ curve for $VPL_{d}$ are higher than the real $\mu_{s}-\mathrm{lg}P(HMI\left|1F\right.)$ curve, i.e., $Slop{e<Slope}_{max}$ and $VPL_{d}<VAL$, which means that RAIM is assessed to be available when using $Slope_{max}$ or $VPL_{d}$. Additionally, both the $\mu_{s}-{\mathrm{lg}}_{a}P(HMI\left|1F\right.)$ curves for $VPL_{c}$ and $VPL_{e}$ are lower than the real $\mu_{s}-\mathrm{lg}P(HMI\left|1F\right.)$ curve, i.e., $VPL_{c}\geq VAL$ and $VPL_{c}\geq VAL$, which means that RAIM is assessed to be unavailable when using $VPL_{c}$ or $VPL_{e}$. Therefore, RAIM availability is conservatively assessed when using $VPL_{c}$ or $VPL_{e}$, and it is successfully assessed when using $Slope_{max}$ or $VPL_{d}$ in this example.

These three specific examples intuitively illustrate that both optimistic and conservative RAIM availability assessments might happen when using the classic-PL-based method; only conservative assessment might happen when using the enhanced-PL method; and both optimistic and conservative assessments can be prevented when using the ideal-PL-based or the slope-base methods.

### 5.2. Simulation Results Statistical Analysis

Both the rates of optimistic and conservative assessments for RAIM availability are calculated for each grid point using the classic-PL-based, enhanced-PL-based, ideal-PL-based and slope-based methods. The optimistic assessment rate is indicated by the ratio between the count of epochs in which RAIM is assessed to be available but is actually unavailable and the count of epochs in which RAIM is actually unavailable. The conservative assessment rate is indicated by the ratio between the count of epochs in which RAIM is assessed to be unavailable but is actually available and the count of epochs in which RAIM is actually available.

Table 4 records the specific optimistic and conservative assessment data for all 4 kinds of RAIM availability assessment methods. As shown in Table 4, both the rates of optimistic and conservative assessments are 0 for each grid point using the slope-based and ideal-PL-based methods. These prove that both optimistic and conservative assessments can be prevented. The reason for the 0 conservative assessment rate of the slope-based method is that the $a_{v}$ step size is small enough (0.01) for offline calculation of $T_{Slope}$, leading to the assigned $T_{Slope}$ value being nearly equal to its true value.

For the classic-PL-based method, there are both grid points with nonzero optimistic assessment rate and grid points with nonzero conservative assessment rate. Figure 11 and Figure 12 present the optimistic and conservative assessment rates for each grid point using the classic-PL-based method, respectively. As shown in Table 4 and Figure 11, there are 2032 grid points with nonzero optimistic assessment rate, 4.65% of the total grid points, and there are 41 grid points with a 100% optimistic assessment rate, meaning that optimistic assessment always happens when RAIM is unavailable at these grid points. As shown in Table 4 and Figure 12, there are 16,678 grid points with nonzero conservative assessment rate, 38.18% of the total grid points, and the maximum conservative assessment rate is 2.59%. Comparing Figure 12 with Figure 11, the coverage area of conservative assessment is much larger than that of optimistic assessment, but the maximum optimistic assessment rate is much higher than the maximum conservative assessment rate for a single grid point using the classic-PL-based method.

For the enhanced-PL-based method, there are no grid points with a nonzero optimistic assessment rate. This finding illustrates that the value of $P_{a}(HMI\left|1F\right.)$ is small enough at $\xi_{b}=\sigma_{m}^{2}\sqrt{\frac{\lambda_{a}}{s_{mm}}}$ for the enhanced PL, which leads to the $\xi_{b}-\mathrm{lg}P_{a}(HMI\left|1F\right.)$ curve always being separated from the $P_{r}(HMI\left|1F\right.)$ line, preventing optimistic assessment. However, there are 43,038 grid points with nonzero conservative assessment rate, as shown in Table 4 and Figure 13, representing 98.53% of the total grid points. The maximum conservative assessment rate is 8.15%. Comparing Figure 13 with Figure 12, the conservative assessment risk using the enhanced PL is much higher than that using the classic PL, which is represented by the much larger conservative assessment coverage area and the higher conservative assessment rate for a single grid point.

According to the above simulation results, the performance of the classic PL is the worst on RAIM availability assessment for both the risks of optimistic and conservative assessments. In particular, an optimistic assessment might cause HMI, which is intolerable. The performance of the enhanced PL is better than that of classic PL because the optimistic assessment is prevented. However, the risk of conservative assessment is significant, reducing RAIM continuity. The performances of the slope and the ideal PL are optimal, preventing both optimistic assessment and conservative assessment.

## 6. Conclusions

According to the theory and simulation analysis, the RAIM availability might be optimistically or conservatively assessed using the classic-PL-based method and might only be conservatively assessed using the enhanced-PL-based method. Using the ideal-PL-based method, both optimistic and conservative assessment can be prevented. However, the calculation of the ideal PL brings a heavy computational burden to the GNSS receiver or the onboard computer. The slope-based method has the same RAIM availability assessment result as the ideal-PL-based method. Because the ideal slope threshold is only related to one geometric parameter, it can be calculated offline and searched online. Thus, the on-board calculation burden can be reduced using the slope-based method. This improved method can be used in RAIM for single GNSS constellation. Further, a semi-physical simulation experiment will be implemented using the common on-board processor to verify the real-time performance of the slope-based method. Moreover, the RAIM availability assessment performance will be verified using massive actual measurement data.

## Figures and Tables

**Figure 1 sensors-24-03283-f001:**
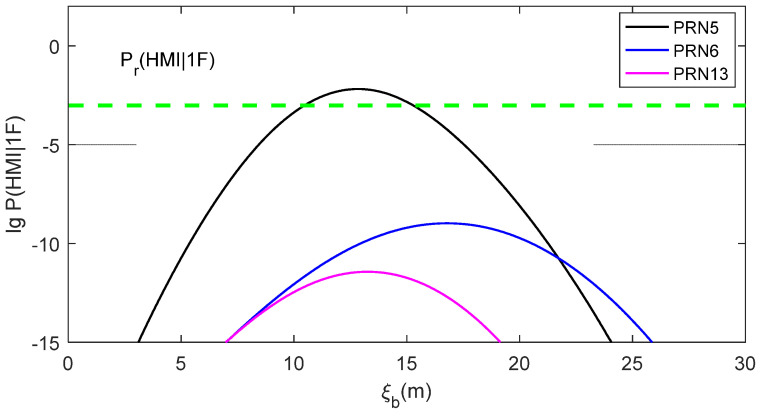
$P(HMI\left|1F\right.)$ for different satellites.

**Figure 2 sensors-24-03283-f002:**
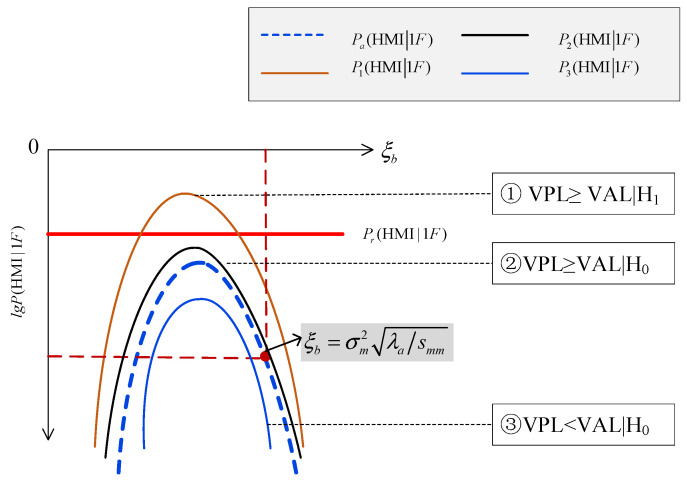
Separation.

**Figure 3 sensors-24-03283-f003:**
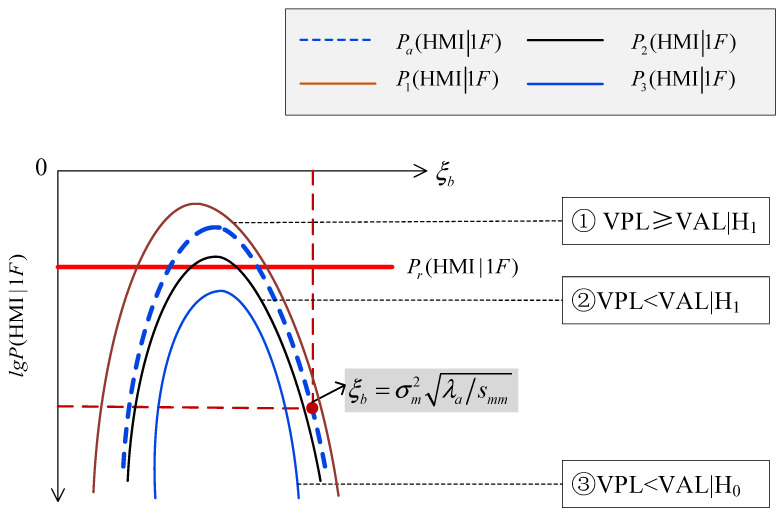
Intersection.

**Figure 4 sensors-24-03283-f004:**
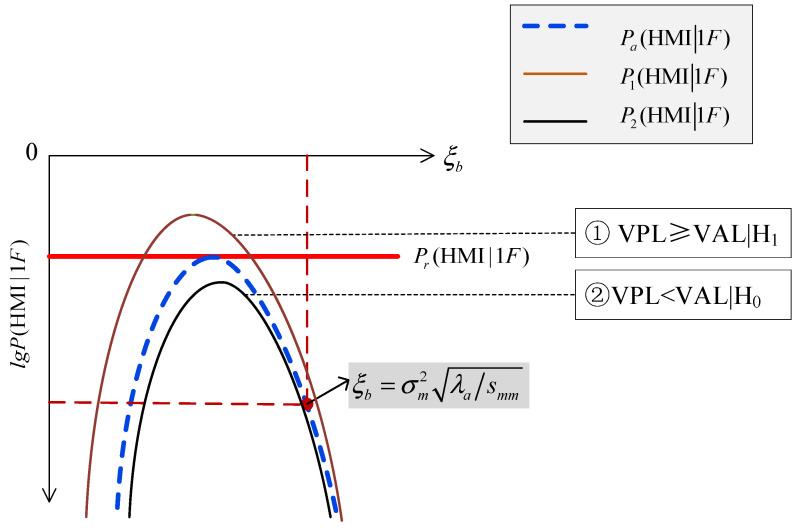
Tangency.

**Figure 5 sensors-24-03283-f005:**
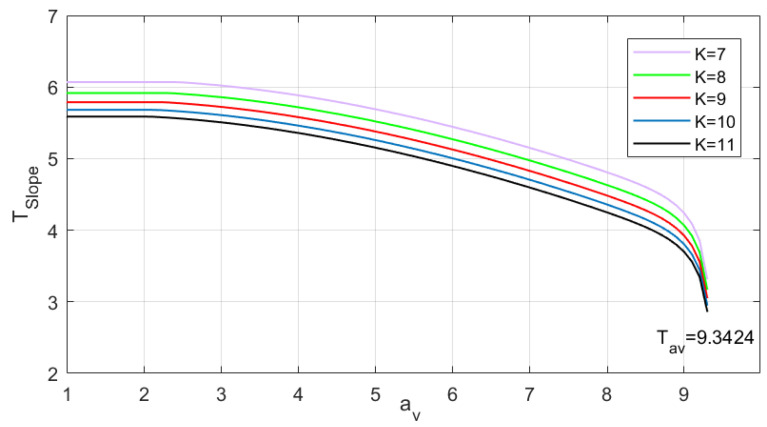
$T_{Slope}$ for different $a_{v}$ values.

**Figure 6 sensors-24-03283-f006:**
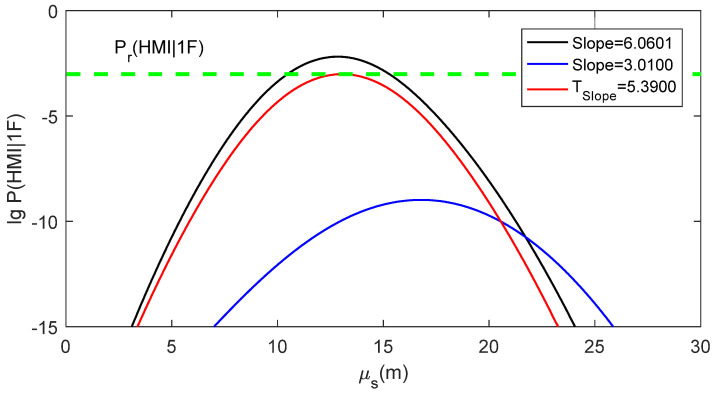
Practical meaning of $T_{Slope}$.

**Figure 7 sensors-24-03283-f007:**
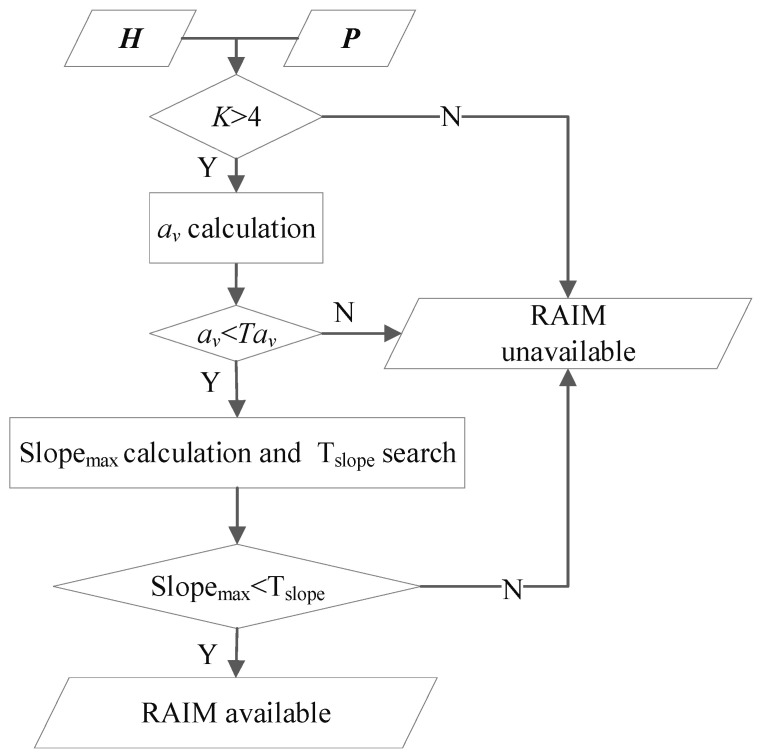
Slope-based RAIM availability assessment process.

**Figure 8 sensors-24-03283-f008:**
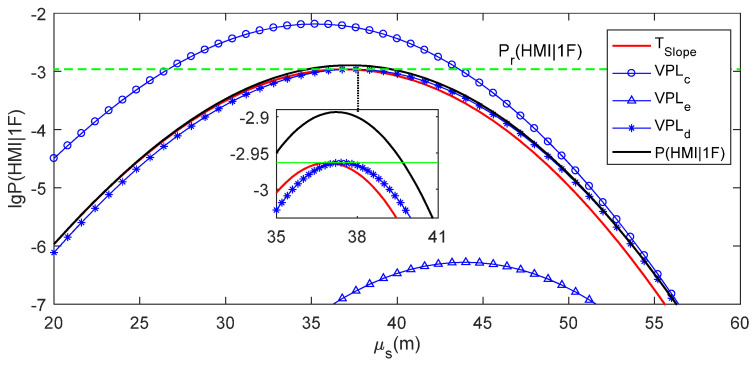
RAIM availability judgment for the first specific example.

**Figure 9 sensors-24-03283-f009:**
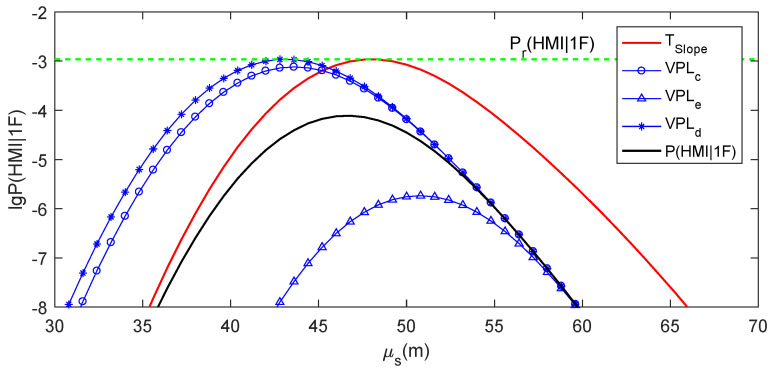
RAIM availability assessment for the second specific example.

**Figure 10 sensors-24-03283-f010:**
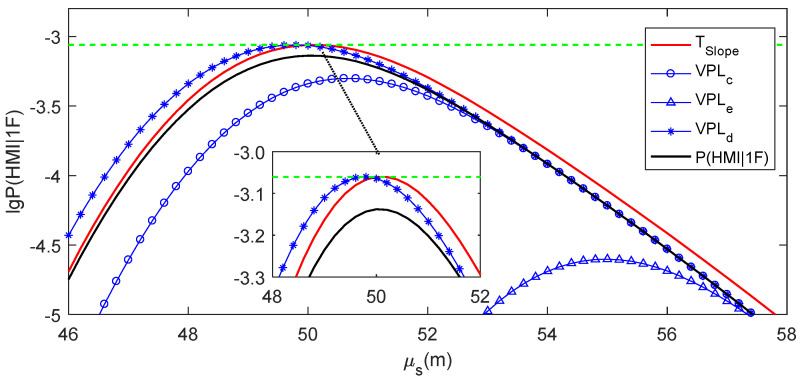
RAIM availability assessment for the third specific example.

**Figure 11 sensors-24-03283-f011:**
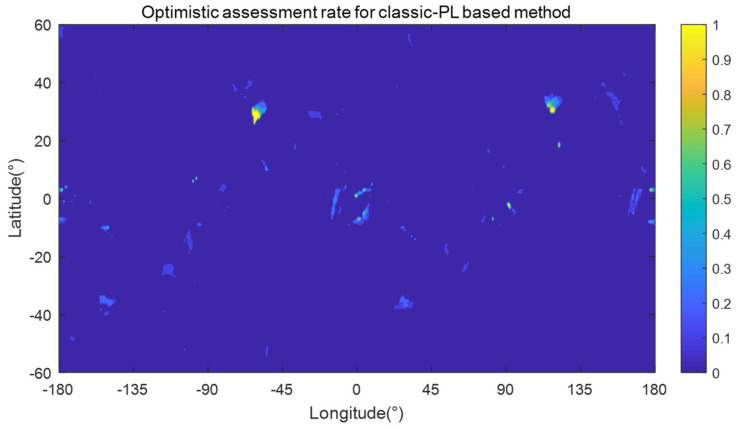
Optimistic assessment rate using the classic-PL-based method.

**Figure 12 sensors-24-03283-f012:**
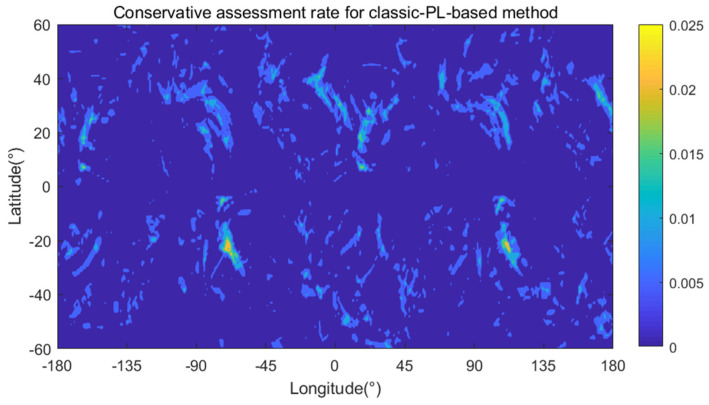
Conservative assessment rate using the classic-PL-based method.

**Figure 13 sensors-24-03283-f013:**
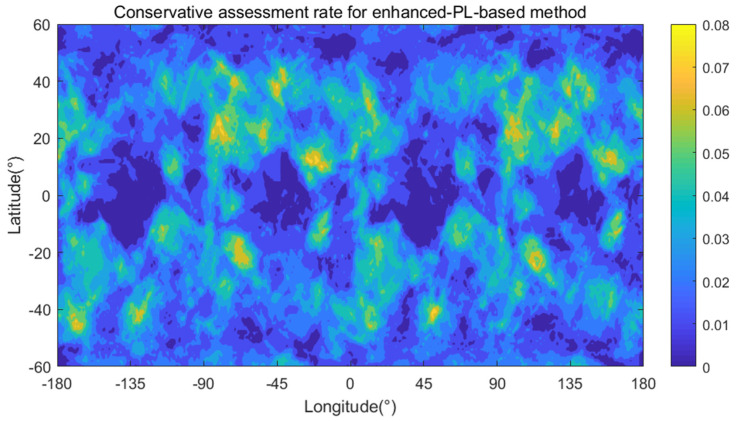
Conservative assessment rate using the enhanced-PL-based method.

**Table 1 sensors-24-03283-t001:** Vertical slope values for different visible satellites.

PRN	2	5	6	12	13	15	19	25	29
Slope	0.162	6.060	3.010	1.576	2.425	1.617	2.036	0.925	1.636

**Table 2 sensors-24-03283-t002:** Step and times of conservative assessment.

Step	0.001	0.01	0.02	0.03	0.04	0.05	0.1	0.2	0.5
Times of conservative assessments	0	0	0	2	2	2	2	2	4

**Table 3 sensors-24-03283-t003:** Specific examples for RAIM availability assessment.

Num	Location	UTC Time	*K*	*a_v_*	*Slope* _max_	VPL*_c_*	VPL_e_	VPL*_d_*	*T_Slope_*
1	8° N, 82° W	03:55:00	8	6.346	5.232	45.287	66.168	50.424	5.172
2	45° S, 132° W	00:00:00	8	2.704	5.330	46.139	55.037	45.475	5.887
3	53° N, 122° E	08:35:00	10	1.358	5.666	50.656	55.305	49.683	5.702

**Table 4 sensors-24-03283-t004:** Performance comparison for the four kinds of RAIM availability assessment methods.

	Item	Grid Points of Optimistic Assessment	Grid Points of Conservative Assessment	Maximum Optimistic Assessment Rate on Single Grid Point	Maximum Conservative Assessment Rate on Single Grid Point
Method					
classic-PL-based		2032	16,678	100%	2.59%
enhanced-PL-based		0	43,038	0	8.15%
Ideal-PL-based		0	0	0	0

## Data Availability

The data used to support the findings of this study are available from the corresponding author upon request.

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
