# Peer review of "An Improved RAIM Availability Assessment Method Based on the Characteristic Slope"

_sensors, 2024, doi:10.3390/s24113283_

Round 1

Reviewer 1 Report

Comments and Suggestions for Authors

The paper is devoted to one of the vital problem of the modern GNSS technologies. Since the GNSS-based technologies became the key stone of numerous modern intellectual transportation, logistic and monitoring systems, the integrity and positioning availability methods and criteria should be considered as the main challenge to achieve the highest GNSS performance. The problem is solved more or less effectively by means of ground and space-based GNSS augmentations in a form of “external” integrity and RNP availability monitoring. But the on-board calculation of correct PL assessments is more complex problem because it needs to be considered, founding an effective availability assessment method which is both the rigor and maintaining the low on-board computational burden. The paper contains important theoretical results allowing to get an effective solution of the abovementioned problem. The subject of the paper is in the scope of the Sensors Journal. Unfortunately the manuscript is written in a messy manner and contains some significant uncertainties which should be corrected before the manuscript acceptance. Hope that my comments attached below will help the authors to improve the manuscript.

Author Response

We would like to thank the reviewer for the constructive comments on this manuscript “A Novel RAIM Availability Assessment Method based on the Characteristic Slope”. The comments are point-by-point replied in the attachment carefully. Please see the attachment.

Reviewer 2 Report

Comments and Suggestions for Authors

1. The RAIM method based on characteristic slopes is not the first of its kind in this paper, and the Novel in the title is questionable.

2. there are too many variables in the paper, which makes it less readable.  Such as  μs-lg P(HMI|1F), ξb-lg P(HMI|1F), etc. Can they be expressed by a noun with some meaning?

3. The description of the formula is not accurate, e.g. 'lg' is an operator rather than a variable. It should not be italicized.

Author Response

(The authors gave the same response as above.)

Reviewer 3 Report

Comments and Suggestions for Authors

The paper is well written and can be nicely followed.

The main work of this paper is analyzing the deficiencies of using classis PL, enhanced PL and ideal PL as the test statistic, then using the characteristic slope as the test statistic and deriving the ideal slope threshold, and making simulation to verify the superiority of the method.

Despite the current research mainstream focusing on ARAIM, due to its small computational burden, RAIM still has practical engineering applications. The proposed method can improve the accuracy of integrity risk assessment for RAIM without increasing the on-board calculation burden.

It is recommended that the authors can further compare the integrity availability assessment results of the proposed method and ARAIM in the future, to provide readers with a more comprehensive reference for applications.

Author Response

Thank you for your recognition of the paper.  We will further compare the integrity availability assessment results of the proposed method and ARAIM .

Round 2

Reviewer 1 Report

Comments and Suggestions for Authors

Dear Authors,

I looked through your responces carefully and saw that all my comments were taken into account accurately. Now I recommend your work to be accepted in a present form.

Thank you for your work